# Experiences from Teaching Practical Machine Learning Courses to Master's Students with Mixed Backgrounds

**Omar Shouman** [1]   **Simon Fuchs** [1]   **Holger Wittges** [1]

## Abstract

*Machine learning education has become more accessible and relevant to students from various backgrounds. Practical courses complement theoretical lectures by focusing on applied machine learning. In this work, we report about our experiences from teaching two machine learning practical courses to master's students from different study programs; an introductory and an advanced course. We present a summary of the teaching and evaluation methods used in both courses. We summarize our experiences and the feedback collected from the students through a survey. We conclude with our recommendations on teaching and designing practical machine learning courses.*

## 1. Introduction

Machine Learning (ML) has recently grown in both relevance and popularity due to its evolving potential in various fields of research. ML technologies are gradually having a significant impact on everyday lives in modern societies (Stone et al., 2016). Because of this, education providers expand their ML-related course portfolio (Engel & Coleman, 2021). In this context, there is a present need for experimented and proven educational methods to teach competences related to ML techniques and tools (Long & Magerko, 2020).

In this paper, we provide a description of two master-level practical ML courses. We also present the methods we used to teach both courses and an evaluation of those methods. In this evaluation, we summarize our experience as teachers and the feedback of the students collected through an on-line survey. We want to share our experience and provide recommendations on the best methods to conduct ML practical courses.

[1]Faculty of Informatics, Technical University of Munich, Germany. Correspondence to: Omar Shouman <omar.shouman@tum.de>.

*Proceedings of the $2^{nd}$ Teaching in Machine Learning Workshop*, PMLR, 2021. Copyright 2021 by the author(s).

## 2. Background

Both ML courses presented in this paper are application-oriented courses (in German, "Praktikum") each worth 10 ECTS (European Credit Transfer System). The courses are offered at the Technical University of Munich to master's students from different disciplines around computer science, information systems, data engineering, and robotics. The first course was designed for advanced learners with a focus on Deep Learning (DL) and took place between October 2020 and March 2021. The second course was an introductory course in applied ML and has been running since April 2021. In both courses, 24 students participated. The students were selected via a matching system that considers both the preferences of the students and the prioritization of the teachers.

The courses were taught completely on-line via Zoom and were split into two main phases; a teaching phase and a project phase. The aim of this organizational split was to allow the students to learn relevant skills in the first part and to apply those skills in a practical project afterwards, where they formed groups of three to four students. The advanced course had a shorter teaching phase (3 weeks) and a longer project phase (11 weeks), while the introductory course had a longer teaching phase (7 weeks) and a shorter project phase (6 weeks). Overall grading and project scope were adapted accordingly.

In both courses, Python was the programming language of choice. Furthermore, we relied heavily on Jupyter notebooks for coding tasks, during the sessions and as homework assignments. In the introductory course, we focused on data science process models such as the Cross Industry Standard Process for Data Mining (CRISP-DM) (Wirth & Hipp), business and data understanding, and the python stack for ML. In the advanced course, the focus was rather on DL projects and the required tools to develop, test, and deploy DL models. The topics included an introduction to common tools such as Pytorch (Paszke et al.), Keras (Chollet et al., 2015), H2O Driverless AI, containerization, and applications of deep learning. Both courses had a module about the ethical dimension of ML and Artificial Intelligence [1].

[1]More details about the courses are available at https://openpower.ucc.in.tum.de/education/practical-courses/.

# 3. Teaching Methods

We briefly discuss the methods we used in the two practical courses. The presented methods are organized into three categories; content delivery, evaluation methods, and feedback channels.

## 3.1. Content Delivery

The sessions were mainly planned for delivering content to the students. Different methods of planning the sessions were employed. For example, block sessions were used to combine theoretical knowledge with hands-on sessions, where the students would listen to a presentation, work on a simple task, and receive feedback afterwards. We utilized such an approach in introducing basics of using scikit-learn (Pedregosa et al., 2011) and in teaching data loading pipelines in PyTorch (Paszke et al.), in the introductory and the advanced courses respectively. Additionally, crash courses were utilized to quickly bring all students to a common level of knowledge. In the introductory course, we held a one-day Python crash course based on ideas and content from (Chan, 2015; Needham, 2020; Severance, 2009). In the advanced course, a two-day crash course on PyTorch was held, since the majority of the audience were already familiar with Keras (Chollet et al., 2015) and/or TensorFlow (Abadi et al., 2016).

From a content perspective, different methods were used to communicate knowledge with the students, motivate interaction, and provide room for discussion. In several sessions, we started with a short presentation with slides. The presentations served as an introduction as well as a warm-up for the topic being discussed. They were used more frequently in the introductory course, iterating over the process of applied machine learning, characteristics and challenges of each phase in the process, and commonly-used tools in the Python stack. The second element we adopted heavily is the use of Jupyter notebooks. The notebooks were made available to the students before class and discussed mostly after the introductory presentations. During the sessions, we coded the Jupyter notebook live and did not walk through the notebook prepared prior to the session. Due to the online-format, we adopted the strategy of alternatively switching between presentation slides and Jupyter notebooks (coding) with a block of 20-30 minutes for each. The target was to overcome the interaction difficulties of the remote setup and connect abstract knowledge with practical use.

We integrated in-class group work to motivate for discussion and enrich the learning experience. The strategy was to divide the students into groups of three to four students and ask them to work on a specific task related to the presented content. The scope of the task was quite versatile, ranging from coding tasks to discussing ideas and brainstorming machine learning solutions. Coding tasks included solving small problems, reading and understanding code, or reading documentation and applying a solution to a different problem or a dataset. After the time dedicated for the task is over, each group briefly presented what they achieved or learned to all other groups and the instructors. When relevant, a short feedback round followed each presentation.

## 3.2. Evaluation Methods

In order to measure the learning progress and give the students the chance to apply the concepts learned during the sessions, students had to work on various tasks as graded homework and project work. In the introductory course, a mini-project covering most of the basic concepts in Python was to be completed. For the advanced course, students had to implement a complete pipeline for a simplified scenario of image inpainting (Zeng et al., 2020) using PyTorch, starting with adapting a dataset for the task, training, and evaluating a simple deep learning model. We used the German traffic signs dataset from (Stallkamp et al., 2012).

Another methodology we used in the introductory course was to provide the students with homework assignments in the form of Jupyter notebooks. The notebooks contained both guided as well as unguided exercises. The guided exercises served the purpose of introducing the usage of libraries such as Pandas (pandas development team, 2020; Wes McKinney, 2010), Matplotlib (Hunter, 2007), and Scikit-learn (Pedregosa et al., 2011). Students had to add their code to the indicated specific parts of the notebook. Later in the course, more unguided exercises were presented, where the students are only provided with a general task formulation with neither code snippets nor structure.

Homework assignments also included a few essay questions to assess the students' understanding of the concepts and their ability to formulate their ideas. Given a specific business context from our research, students of the introductory course were asked to identify use-cases for machine learning, motivate them, and prioritize them according to their business impact. For the project work, we opted for a high-level project scope, where groups of three to four students were asked to develop a concrete proposal with their ideas and plans. The requirements were to adhere to a set of milestones and deliverables, while providing room for the students to extend the project scope, integrate auxiliary modules in their implementation, and explore new ideas. The project for the introductory course involved developing a complete solution using machine learning for an actual business use-case based on an internal dataset we curated from a running system. The project of the advanced course was to develop a face recognition pipeline using existing state-of-the-art deep learning models [2]. Students from the advanced course used GPU resources provided by IBM to

---

[2]Project reports and code are publicly available here.

train and fine-tune their deep learning models.

### 3.3. Feedback Channels

We believed feedback is an integral component of the learning process; therefore, we adopted multiple feedback channels, where each channel is dedicated to a specific scope or element of course. Table 1 lists the channels we used and the corresponding scopes of questions.

*Table 1.* Feedback channels and corresponding scope of questions.

| QUESTION SCOPE | CHANNEL |
| --- | --- |
| GENERAL | CHAT PLATFORM AND FORUM |
| HOMEWORK ASSIGNMENTS | Q&A LIVE SESSIONS BEFORE SUBMISSION
TEXTUAL COMMENTS AFTER SUBMISSION |
| PROJECT-RELATED | WEEKLY OFFICE-HOURS
FEEDBACK AFTER PRESENTATIONS |

## 4. Experiences

We report about our experiences from both courses, focusing on three major aspects; methods of content delivery, scope of tasks, and project work.

### 4.1. Content Delivery

We had a positive experience with integrating a mixture of methods in the same session when delivering content to the students. Concretely in the case of teaching practical machine learning, we slowly arrived at the following sequence of teaching activities in our sessions: short presentation with slides, live-coding session, group work, and finally presenting and discussing with all groups. Live-coding in an empty Jupyter Notebook worked out better than going through the notebook and executing the cells. Despite being more time-consuming, we found out that it improved the engagement and the follow-up of the students by regulating the pace of presenting and developing ideas. We also found that using a running use-case along several modules makes it easier for students to follow up and connect the different topics, we were inspired by the end-to-end machine learning project chapter from (Géron, 2019). Due to the practical nature of both courses, we designed and delivered the content following a suitable process model; CRISP-DM for the introductory course and a more DL-specific process model adopted from (Raghu & Schmidt, 2020) for the advanced course. This turned out to be useful in understanding the holistic overview of the iterative process and logically connecting the various steps.

### 4.2. Scope of Tasks

When scoping tasks for the students, we found out that realistic scenarios involving ambiguity provide a better learning opportunity for students. They simulate real-life ML problems and enable students to stretch their thoughts beyond standard toy examples. They also touch upon important skills such as identifying possible use-cases for ML given a complex business scenario, formulating each identified use-case correctly, and validating assumptions based on the available data. However, they come at the cost of being more challenging and time-consuming for both the teacher and the student. For the more practical phases of the ML process such as learning how to use a package, guided exercises proved very successful as a first step that can be later complemented with unguided exercises. Although unguided exercises are relatively challenging, they represent a more realistic scenario allowing the students to develop their own work and tackle the problem systematically.

### 4.3. Project Work

From our experience, a flexible project scope has increased the motivation of the students. They formulated major parts of the project by themselves and demonstrated full-ownership of the whole work. Some groups explored new ideas, complemented the suggested pipeline with more tasks, and made demos for their implementations. When forming the groups, we found out that heterogeneously mixing them with respect to background engages all students and evenly distributes workload. During the project phase, we realized the importance of milestones, where the students can present their work and get constructive feedback. As explained, this was conducted in the form of intermediate presentations and regular office-hours, where meetings were held with each group separately.

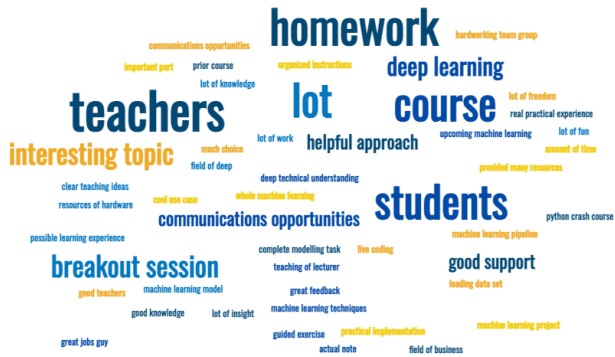

*Figure 1.* Wordcloud of the textual responses of the students.

## 5. Student Feedback

Course participants were asked to evaluate several aspects of the course through an on-line survey. The survey consisted mainly of multiple-choice questions along with two essay questions where the students can deliver further feedback. The response rates for the introductory and the advanced course were 50% and 38%, respectively. Although the sample size is relatively small, it indicates a general trend of

the experience of the students. The feedback of the students from the essay questions is summarized in a wordcloud in Figure 1.

## 5.1. Overall Learning Success

In the survey, students were asked to self-assess their skills in machine learning before and after the course. Students could respond on a 5-point Likert scale (1 = not at all; 5 = very much). The averages for the introductory course and the advanced course moved from (2.4, 2.9) to (3.7, 4.2), with a difference of 1.3 points. Additionally, all students assessed their skills with a higher score after the course than before.

## 5.2. Best-Evaluated Teaching Methods

On the same Likert scale, students evaluated the different teaching methods we used during the course sessions. The top five methods in both courses are shown in Table 2.

Clearly, the project-work and coding assignments were the top-rated methods. Other methods such as "Group-work during the sessions", "Homework essay questions", and "Literature recommendations" were graded with lower average scores; 3.9, 3.3, and 2.9, respectively.

*Table 2.* Top-5 methods evaluated by the students and their average score on a 5-point scale (1 = not helpful; 5 = very helpful).

| Method | Introductory | Advanced |
| --- | --- | --- |
| Working on the project | 4.75 | 5.0 |
| Learning from exemplary code | 4.83 | 4.75 |
| Coding homework | 4.66 | 4.88 |
| Office hours & Individual discussions | 4.36 | 4.83 |
| Slide presentations via Zoom | 4.25 | 4.5 |

## 5.3. Group-Work and Individual-Work

Another interesting outcome of the survey was that the students evaluated individual learning consistently higher than group-work, except for the course project. To put these results in context, all in-class group-work activities were based on random assignment of group members via the on-line conferencing software. However, students had the chance to work together within the same group for an extended period of time on the course project. Since both courses were conducted remotely, the lack of social interactions among the groups can have an impact on such results, especially when groups are temporarily formed during on-line sessions.

## 5.4. Crash Courses

Since we used crash courses to teach practical skills at the start of each course, students were also asked to evaluate

them. On the 5-point scale, students from the introductory course evaluated the Python crash course with an average of 3.8, taking into consideration that 40% of the participants were previously familiar with Python. For the advanced course, participants evaluated the PyTorch crash course with an average of 4.8, where only 10% of the participants used it at least once before the course.

## 6. Conclusion

In this paper, we present teaching methods used in two practical ML courses. We also summarized our experiences as teachers with both courses; and the feedback from the students collected via a survey. We derive recommendations for teachers on the methods to use for planning the sessions, delivering content, designing assignments, and choosing project-work. A summary of our recommendations is presented in Table 3.

*Table 3.* Summary of recommendations for practical machine learning courses.

| Teaching | Evaluation & Feedback |
| --- | --- |
| Crash Courses & Block Sessions (to level up skills) | Coding Homework (mix guided & unguided) |
| Slide presentations (concise as introduction) | Mini-projects (include complete ML pipelines) |
| Jupyter notebooks (live-coding) | Project scope (flexible, real-world scenarios) |
| Exemplary code (well-written and documented) | Project groups (mix with respect to backgrounds) |
| In-session Group work (focus on coding) | Regular feedback (also during project phases) |

In our experience and according to the students' feedback, practical coding tasks based on realistic use-cases are successful methods for teaching machine learning. Additionally, teaching techniques that involve live-coding, either led by the instructor or done as in-class group work, have contributed to a better learning experience. Jupyter notebooks provide a flexible environment for learning; however, they can result in the inability to work outside them. This is a challenge that requires further investigation. For coding assignments, the combination of guided and unguided exercises trains the students to progress from simple tasks to more advanced and complex ones. Crash courses and block sessions level up the knowledge of the students to a common level, addressing the challenge of teaching a diverse student body. Finally, projects provide a great learning opportunity for students, given that they are complemented with regular feedback sessions and concrete milestones.

## Acknowledgements

We are grateful to IBM for providing the students with hardware resources to use during both the block sessions and the projects. We would like to thank team members from IBM Germany for their valuable contribution to the block sessions of the advanced course and for delivering a guest talk during the introductory course.

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
