# OpenReview forum: "Experiences from Teaching Practical Machine Learning Courses to Master's Students with Mixed Backgrounds"
_ecmlpkdd.org/ECMLPKDD/2021/Workshop/TeachML — TeachML 2021_

### Official Review · Reviewer_w3XW · 2021-07-15
**Good paper about teaching ML practices to diverse students**

**Rating:** 9
**Confidence:** 4

**Review:**

This paper is an experience report from teaching two machine learning courses, an introductory one and an advanced one, during the current SARS-CoV-2 pandemic, in an online format.

The paper is a great read, it is very well written, and contains very good content and advice for teaching machine learning courses, specially about multiple ways to deliver content and obtain student feedback.

I definitely agree that students enjoy live coding much more than just executing some pre-defined code, I think this paper adds to the evidence that this method should be used more often.

The paper is closed with a survey evaluation of most aspects of both courses, which is quite helpful for future teachers as it contains evidence of which practice is preferred by machine learning students, and future best practices for teaching courses during the pandemic, and also in in-person formats.

I have only three comments about improvement of this paper:

- First, I think should be considered, is that while jupyter notebooks are very good for teaching, sometimes the students can fail to generalize while using them, being unable to work outside of a jupyter notebook environment (for example in a python shell). This comes from my own observation and experience, so some comments about this could be useful, but this might require some more long term evaluation.
- Second, there seems to be little connection between the argumentation in the paper and teaching for a diverse student body, I think the authors could strengthen this point, I see that most of the teaching methodology implicitly it is tailored for mixed background students, but some comments on how they decided on these techniques and connect them to their student body, would greatly improve the paper.
- Finally, I think adding the course contents or short syllabus as part of the appendix/supplementary material would also help other teachers in the future.

---

### Official Review · Reviewer_zVLn · 2021-07-16
**Interesting insights about teaching applied ML to master students**

**Rating:** 8
**Confidence:** 4

**Review:**

The paper is well written and clear. It describes an applied machine learning course for master students.

Pros
- student feedback is included and positive
- technological and software tools used are described
- paper includes clear recommendations for applied machine learning courses

Cons
- the machine learning content given in the course is not detailed.

---

### Decision · Program_Chairs · 2021-07-21

**Decision:**

Accept

**Comment:**

Congratulations! The reviewers agree that this paper should be accepted.

Camera-ready version is due August 18, 2021. As you prepare the camera ready version, please take the reviewers comments into consideration.

We look forward to your participation at the workshop on September 13, 2021. We invite you also to join us for the satellite event on September 08, 2021. Schedules for both the workshop and the satellite event will be forthcoming.